

# The effect of conventional and sustainable agricultural management practices on carbon and water fluxes in a Mexican semi-arid region

Gabriela Guillen-Cruz[1,*], Roberto Torres-Arreola[1,*], Zulia Sanchez-Mejia[2] and Dulce Flores-Renteria[3]

[1] Departamento Sustentabilidad de los Recursos Naturales y Energía, Centro de Investigación y de Estudios Avanzados del IPN, Unidad Saltillo, Ramos Arizpe, Coahuila, Mexico
[2] Departamento de Ciencias del Agua y Medio Ambiente, Instituto Tecnologico de Sonora, Ciudad Obregon, Sonora, Mexico
[3] Conacyt-Sustentabilidad de los Recursos Naturales y Energía, Centro de Investigación y de Estudios Avanzados del IPN, Unidad Saltillo, Ramos Arizpe, Coahuila, Mexico
[*] These authors contributed equally to this work.

Corresponding author
Dulce Flores-Renteria,
yaahid.flores@cinvestav.edu.mx,
yaahid@gmail.com

## ABSTRACT

**Background**. Agriculture is essential for food security. However, conventional agriculture alters the water and carbon cycle and soil properties. We investigated the effect of conventional management (CM) and sustainable management (SM) on the carbon and water cycle in crops of nopal (Np) and wheat (Wh).

**Methods**. A micrometeorological eddy covariance tower was installed to measure water use through evapotranspiration (ET) and the net exchange of $CO_2$ during the crop's development. Gross primary productivity (GPP), water use efficiency (WUE), and soil properties were obtained.

**Results**. The results showed that both agricultural managements influenced the carbon flux of the ecosystem, with a lower GPP and Reco in the nopal field (1.85 and 0.99 mmol C m$^{-2}$ s$^{-1}$, respectively), compared to the wheat field (6.34 and 1.8 mmol C m$^{-2}$ s$^{-1}$, respectively). It was mainly attributed to the metabolic plant differences, phenological stages, and wheat biomass developed during the winter. On the other hand, the accumulated ET in the SM-Wh plots was lower than SM-Np. Therefore, the crops subjected to sustainable practices use water more efficiently with 1.42 and 1.03 g C m$^{-3}$ $H_2O$ for nopal and wheat, respectively. In regard to soil properties, it was observed that tillage alters microbial activity affecting organic matter and carbon. It can be concluded that the differences in agricultural management for both crops altered the carbon and water cycle and soil quality. In addition, implementing good agricultural practices allows more efficient use of water by the plant, higher retention of water in the soil, and less ET.

## INTRODUCTION

Agricultural activities encounters several challenges including water consumption, and waste (*D'Odorico et al., 2020*), greenhouse gas emissions like $CO_2$, $N_2O$ and $CH_4$ (*Friedlingstein et al., 2020*; *Mirzaei & Caballero Calvo, 2022*; *Mohammed et al., 2022*). Currently, there are different agricultural management techniques; however, the conventional (*e.g.*, monocultures, deep plowing, application of synthetic chemicals, genetically modified organisms) approaches are ubiquitous worldwide and in Mexico (*INEGI, 2019*; *Mirzaei et al., 2021*). Furthermore, due to the adverse impacts of conventional agricultural practices on carbon and water cycles (*Davis et al., 2010*; *Fisher et al., 2017*; *Camarotto et al., 2018*), and on the soil quality (*Nadeu et al., 2015*), it is crucial to promote the use of sustainable agricultural practices. In addition, due to the effects of climatic and edaphic characteristics on sustainable practices, it is necessary to know how they affect locally to determine the feasibility of application at this scale.

The Food and Agriculture Organization (*FAO, 2020*) has recommended a series of "Good Agricultural Practices" (GAP), which are based on the maintenance of soil organic carbon (SOC). These are: (i) management of available water and efficient irrigation; (ii) reduced plowing, minimal or no plowing or residue management; (iii) maintenance of pastures to maintain vegetative cover; (iv) use of cover crops (a close-growing crop that provides soil protection, seeding protection, and soil improvement between periods of normal crop production), perennials, or pastures; (v) balanced use of fertilizers or use of organic amendments (compost, animal manure, plant residues); (vi) the use of biofertilizers; (vii) crop rotation and the use of improved species; (viii) integration of production systems (silvopasture, agroforestry); (ix) landscape management to prevent erosion, surface water management, and drainage; (x) the cultivation of indigenous plants. These practices aim to conserve soil health and reduce water consumption. Measuring the different management practices' effect on carbon and water fluxes and the soil quality at a local scale will permit a contrast between conventional and GAP (*i.e.*, i, ii, and v) to reduce water consumption, avoid carbon emissions to the atmosphere, and conserve the soil. These practices are much needed in semi-arid regions where the human population is increasing, and where a better use of resources for the food supply needs to be considered.

Carbon and water fluxes in agricultural systems have been the subject of several studies (*Xiao et al., 2011*; *Yang et al., 2017*; *Cleverly et al., 2020*; *Wagle et al., 2021*; *Mirzaei & Caballero Calvo, 2022*). One of the most widely used technique for measuring gaseous exchanges between the ecosystems and the atmosphere is eddy covariance (EC) (*Baldocchi, 2014*; *Kautz et al., 2019*; *Baldocchi, 2020*; *Thienelt & Anderson, 2021*). Measurements using EC can help elucidate the temporal and spatial variability of energy, water and $CO_2$ fluxes and the conditions that affect them, such as the different agricultural management (*Niu et al., 2011*; *Vote, Hall & Charlton, 2015*). At ecosystem scale water is loss *via* evapotranspiration (ET), and $CO_2$ can be gain or loss via photosynthesis and respiration respectively (*Baldocchi, 2020*). Both processes are coupled through the water use efficiency (WUE), or ratio between C assimilation per water used (*Cai et al., 2021*), which can be estimated with GPP or crop yield. Carbon and water fluxes have been studied on wheat, a

staple cereal in the human diet (*Guan et al., 2020*) and more recently cactus crops (Nopal), as the latter are adapted to arid areas (*Consoli, Inglese & Inglese, 2013*; *López Collado et al., 2013*; *Guillen-Cruz et al., 2021*). The study of these fluxes is important for achieving sustainable agricultural goals and to improve WUE. In addition, the carbon and water cycles of agricultural systems are influenced by location, soil type, and management practices such as fertilizer application, type of irrigation, crop rotation and selection (*Drewniak et al., 2015*; *Waldo et al., 2016*).

The objective of the study is to evaluate the effect of conventional and sustainable management over the soil quality, and the carbon and water fluxes. We hypothesize that (a) the sustainable management would show a better water use efficiency and a higher soil quality, due to the conservation of resources that these practices promote; and (b) using a native crop (nopal) would show a low carbon and water fluxes, due to the adaptations of this plants to the dry conditions, in comparison with no-native crops, like wheat. Therefore, native crops like nopal would present better WUE than introduced crops.

## MATERIALS & METHODS

### Study site description

The study site was established within the San Isidro ranch at the General Cepeda municipality of Coahuila de Zaragoza state in northern Mexico (Fig. 1). Climate is semiarid BSKw (*García, 2004*) with a mean annual temperature of 18.4 °C; the hottest month has a maximum temperature of 30.3 °C, and in the coldest month, a minimum temperature of 6 °C. The mean annual precipitation is 377 mm, with the highest precipitation during July, August, and September (*CICESE, 2018*).

The dominant vegetation is microphile and rosetophile desert scrub, with dominant species such as *Fouquieria splendens*, *Larrea tridentata*, *Yucca carnerosana*, *Y. filifera*, *Dasylirion cedrosanum*, and various species of the Cactaceae family (*Campuzano, Delgado-Balbuena & Flores-Renteria, 2021*; *Flores-Rentería et al., 2022*). The main crops established in the municipality are corn, beans, and wheat, with rain-fed monoculture agriculture. The area used to establish experimental plots was abandoned for approximately 20 years.

### Experiment design

The study area was divided into four 10 × 10 m plots. Two plots were planted with nopal (*Opuntia ficus-indica* L.), and the two other plots with local variety wheat (*Triticum aestivum*, var. AN373F2016) adapted to the region. Each crop was given conventional or sustainable management with a factorial design (Fig. 1). For the conventional management (CM), conventional tillage was applied by plowing to depth of 30 cm, regardless of the crop; then a crawl was carried out with discs, breaking large clods of soil, finally grooves were used to even soil surface. The wheat conventional management also included chemical fertilization with 2 g m$^{-2}$ of liquid urea on two periods, 22 and 158 days after sowing (DAS). In the case of sustainable management (SM), minimum tillage was applied for the wheat crop only in the top 10 cm of soil depth, in addition, a seeder was used in the wheat plots. In the case of nopal, no tillage was applied, only individual holes were dug. Organic fertilization was applied at 1 kg m$^{-2}$ of pork compost. The wheat sowing was

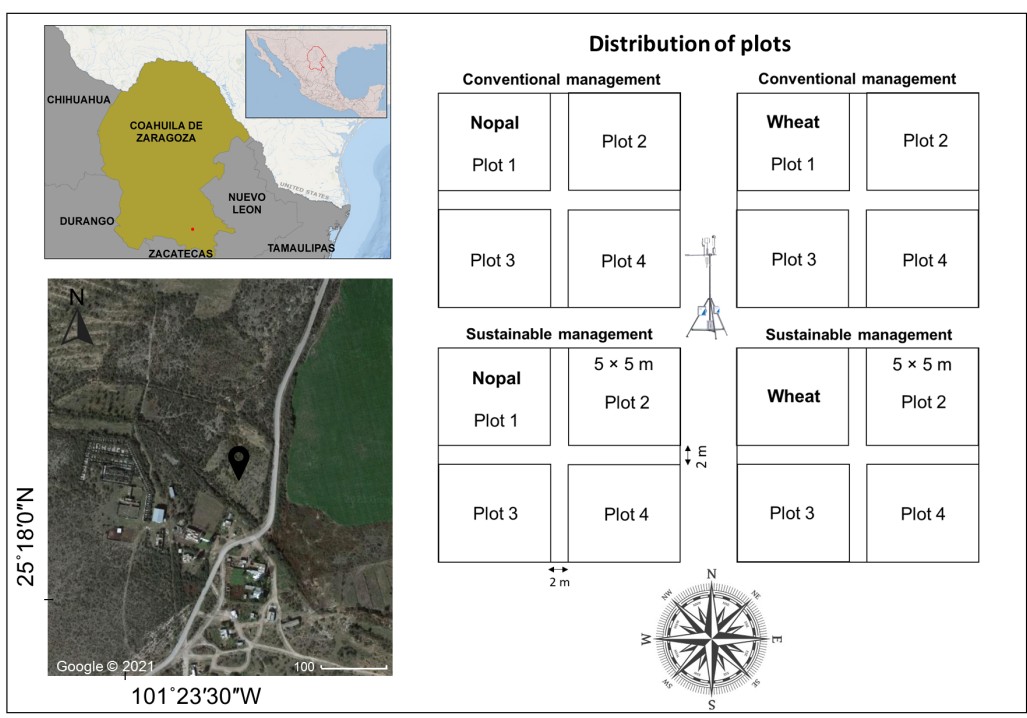

**Figure 1** Site location and distribution of experimental plots with Nopal and Wheat crops subjected to conventional or sustainable practices in a Mexican semi-arid region. Map data © 2022 Google.

done with a planter (80 kg of seed per h), whereas nopal (20,000 per h) was half-buried manually, regardless of the management. Wheat (Wh) was sowed in early September 2020 and harvested from plots with CM in April 2021 (236 DAS) and WH with SM in May 2021 (263 DAS). Nopal (Np) crop was established in early September 2020. The nopal growth nonetheless of the management was insufficient to harvest, since did not reach the minimum of two cladodes levels or cladodes 15–25 cm length (*Flores-Valdez, 1995*). According to regional recommendations, harvest before reach this minimum size would not reflect the real production values (*Luna Vázquez, 2011*).

Two 450-liter water tanks connected to a water pump were installed for better water pressure. A meter was also installed to keep track of irrigation. Nopal (Np) plots were drip-irrigated, and a micro-aspersion system irrigated the Wh field. The amount of water applied was controlled weekly according to the crop requirements. Irrigation applied for the entire growth period was 610.30 mm for SM-Np, 738.99 mm for CM-Np, 100.20 mm for SM-Wh, and 173.61 mm for CM-Wh. Average soil water content (SWC) monitored continuously with WatchDog sensors and 1,400 datalogger (Spectrum Technologies, Inc., El Paso, TX, USA). SWC was maintained for Wh between $8 \pm 4\%$ and in $17 \pm 5\%$ by applying between 5–20 $mm^3$ for SM and 15–45 $mm^3$ for CM (*Avila Miramontes et al., 2014*). In the case of Np SWC was maintained between $8 \pm 5\%$ and in $5 \pm 4\%$, according to regional recommendation and based on different studies (*Luna Vázquez, 2011*; *FAO, 2018*). Irrigation was maintained along the crop period, except in the case of end-Wh, where the

irrigation was stopped to permit grain maturation. The accumulated precipitation for the crop period was 126.7 mm (Fig. S1).

## Soil and plant properties

Soil cores (two cm in diameter) were taken from a depth of 0–15 cm before tillage (pre-tillage), after tillage (post-tillage), and after harvesting (post-harvesting) at each plot ($n = 3$), for a total of 12 samples per treatment. Soils were sieved (<2 mm) and air-dried. Soil pH was determined in a 1:2 (w:v) aqueous suspension (Orion Star A211, Thermo Fisher Scientific, Inc, Waltham, MA, USA). Soil quality was evaluated by its soil physicochemical characteristics: soil organic matter (SOM) was assessed by loss on ignition at 400 °C for 4 h. Organic carbon (COS) was determined by Walkley and Black method (*Walkley & Black, 1934*). Available phosphorus and nitrogen were determined using Bray's and Kjeldahl's method (*Bray & Kurtz, 1945*; *Bremner, 1960*). The volumetric method determined the soil bulk density. The soil texture was determined according to Bouyoucos hydrometer (ASTM 152H) (*Ashworth et al., 2001*).

After cropping, Wh crop was characterized by taking five cores of the plant crop (4″ diameter) at each plot. For the assessment of aerial wheat biomass, plant material was dried in a forced-air oven at 60 °C to a constant weight, weighed, and separated into grain and straw. Quality of the crop was characterized by the analyses of nutrients in the aerial part of the plant, since the main purpose of the Wh variety is as fodder. For this, nitrogen content by Kjeldahl's method ($H_2SO_4$ digestion) (*Bremner, 1960*); phosphorus content by vanadate/molybdate method (*Hanson, 1950*); potassium content by atomic emission (Thermo Scientific, Model ICE3300); and Fe content by atomic absorption (Thermo Scientific, Model ICE3300). For the P, K, and Fe determinations, the plant samples were first digested in $HNO_3 + HClO_4$ 2:1 solution (*Huang & Schulte, 1985*).

## Soil respiration

To determine the soil respiration (Rsoil) rate, a portable dynamic closed chamber type SRC-1 (PP Systems, Amesbury, MA, USA) was used, attached to an infrared gas $CO_2$ analyzer (EGM-5; PP Systems, Amesbury, MA). Rsoil measurements (60 s) were carried out between 12:00–15:00 h at maximum daily soil temperature (*Campuzano, Delgado-Balbuena & Flores-Renteria, 2021*) in three points per plot for each crop and management ($n = 12$). Rsoil measurements were made by synchronizing the irrigation of the two crops with the different agricultural managements, approximately every 50 days and before and after irrigation. At each sampling point and during the Rsoil measurement, soil moisture and soil temperature was measured with a sensor (Hydra Probe II; Stevens Water Monitoring Systems, Inc., Portland, OR, USA) connected to the EGM-5, this was introduced into the soil at eight cm depth. In addition, a micro station (WatchDog 1450; Spectrum Technologies, Inc., El Paso, TX, USA) was used to measure relative humidity and ambient temperature. After that, photosynthetically active radiation was measured with a MQ-200 sensor (PAR; Apogee Instruments, Logan, UT, USA).

## Meteorological and net ecosystem exchange measurements

At the center of the experimental plots (25°18′07″ N, 101°23′24.01″ W), a 3 m micrometeorological tower was installed. An eddy covariance system conformed by a three-dimensional sonic anemometer (WindMaster Pro, Gill Instruments, Lymington, UK) and an open path infrared gas analyzer (LI-7500DS; LI-COR Biosciences, Lincoln, NE, USA) was used. Flux data were sampled at 10 Hz and stored in a USB device in the SmartFlux® 3 system, averaging files every 30 min along the crop period from September 3rd, 2020, to April 27th, 2021.

Additionally, meteorological variables were continuously collected during the same period at a rate of five seconds and averaged and stored at half-hour intervals using a datalogger (Sutron Xlite 9210). Both a quantum sensor (LI-190R-SMV-5, LI-COR Biosciences) and a pyranometer sensor (LI-200R-SMV-5; LI-COR Biosciences) were used to measure the photosynthetic active radiation (PAR); whereas the net radiation was measured with a radiometer (NR-Lite2; Kipp & Zonen). A Vaisala sensor (Vaisala HMP155) was used to measure the relative humidity (RH) and air temperature (Tair). Soil heat flux was monitored by three soil heat flux plates (HFP01; Hukseflux Thermal Sensors BV) at eight cm depth. Three soil moisture and temperature probes (Hydra Probe II, Stevens) at five cm depth were also placed. A tipping bucket rain gauge (TE525, Texas Electronics) was installed to monitor precipitations.

## Data processing

Raw eddy covariance data were processed in EddyPro®, as previously described in *Campuzano, Delgado-Balbuena & Flores-Renteria (2021)*; *Flores-Rentería et al. (2022)*. Specifically, night time fluxes below the threshold of $u\star = 0.12$ m s$^{-1}$ were removed; this threshold was defined through the 99% criterion (*Reichstein et al., 2005*). Additionally, 90% of cumulative fluxes $<= 40$ m, footprint model was retained (*Kljun et al., 2004*). In the data period, 56% of half-hour data were lost after quality filtering. The energy balance closure was >90% for the whole period.

Since the plots were well oriented towards each cardinal point, the peak fluxes separation was possible using MATLAB R2021a (MathWorks, Inc., Natick, MA, USA). The footprint data was obtained for the entire growth period, and then the fluxes were separated by management and crop, according to the cardinal point where the data were obtained. The data fluxes were separated into angular sectors of the wind direction. At $\theta < 90°$ for CM-Wh, $90° < \theta < 180°$ for SM-Wh, $180° < \theta < 270°$ for SM-Np and $270° < \theta$ for CM-Np (Fig. 2). After quality filtering and separation of data we had 42.3% of the half-hour data from CM-Wh (5,698), 14.6% from SM-Wh (1,962), 21.6% from SM-Np (2,909) and 21.5% from CM-Np (2,893). Flux partitioning was performed in the online MPI Jena tool (http://www.bgc-jena.mpg.de/REddyProc/brew/REddyProc.rhtml), using Tair by the night-time-based flux-partitioning algorithm (*Reichstein et al., 2005*).

## Data analysis

The conversion factor of 0.1584 (*Lamptey, Li & Xie, 2018*) was used to convert the Rsoil rates from g $CO_2$ m$^{-2}$ h$^{-1}$ to μmol $CO_2$ m$^{-2}$ s$^{-1}$. Water use efficiency (WUE) was calculated

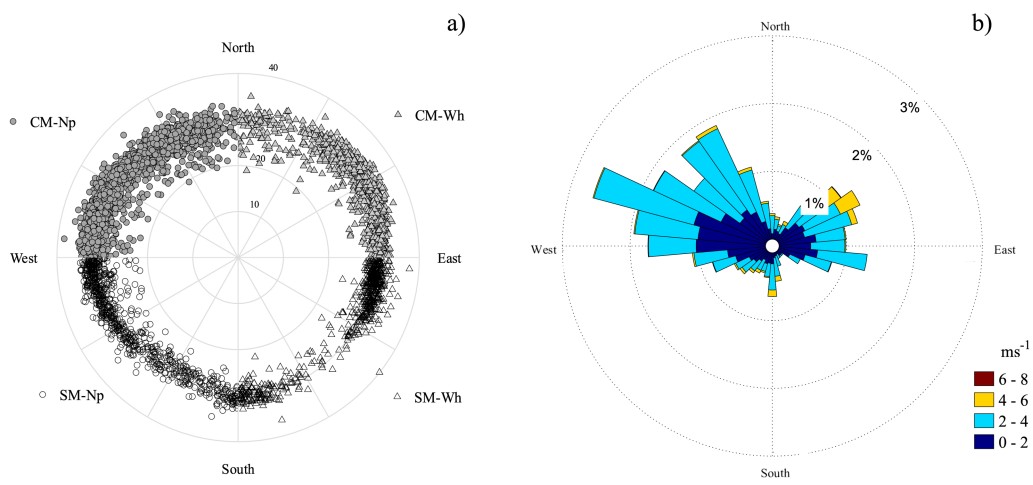

**Figure 2** **(A) Footprint and (B) wind rose of crop data.** Northeast conventionally managed wheat; southeast sustainably managed wheat; southwest sustainably managed cactus; and northwest for conventionally managed cactus. Conventionally managed wheat (CM-Wh; gray triangle, $\theta < 90°$); sustainably managed wheat (SM-Wh; white circle, $90° < \theta < 180°$); sustainably managed cactus (SM-Np; white circle, $180° < \theta < 270°$); conventionally managed cactus (CM-Np; gray circle, $270° < \theta$). The direction of the stripe shows the wind direction, and the color of the stripe indicates wind speed (m s$^{-1}$) (Sep 3th–May 26th).

as the ratio of GPP to ET (*Cai et al., 2021*), where GPP was obtained by partitioning NEE and ET with the EC method: WUE = GPP/ET. For the calculation of WUE, the average GPP in the growing season and accumulated ET of the same period were taken.

One-way analyses of variance (ANOVA) ($p < 0.05$) were performed to assess the effect of management for a given crop on soil and plant properties, $CO_2$ fluxes, and evapotranspiration. Tukey's honesty test was used to evaluate significant differences between means. All ANOVAs were run in R (*R Core Team, 2020*). Normality and homoscedasticity of the residuals was met in all analyses. To investigate the variable importance between environmental factors and carbon fluxes in the experimental plots, Principal Component Analysis was used, using *princomp* function in R (*R Core Team, 2020*).

## RESULTS

### Effect of agricultural management on soil and crop yield

The initial soil texture in the experimental plots was clay loam with proportions of sand, silt, and clay 34:35:31. Initial tillage and fertilization modify the soil's organic matter, organic carbon, and bulk density (Table 1). Specifically, the application of 30 cm-tillage positively impacted the organic matter content top layer. This tillage also incremented the bulk density, with lower values shown in CM in both Np and Wh. In the case of soil chemical properties, the soil pH and electric conductivity were also affected by tillage; conventional tillage (30 cm) results in higher pH and lower electric conductivity than minimum tillage (10 cm). The post-harvesting soil quality (*i.e.*, SOM, pH, EC, nutrients) also reflected the

effect of management in a short period (one crop cycle). This effect was evident in the case of the Wh crop, whereas the Np crop did not show significant differences in the soil quality associated with the management. The soils from SM-Wh presented higher bulk density, pH, and organic matter content, as well as higher available N and P.

Only Wh crop yield and plant nutrients were characterized due to the short evaluation period for Np (263 DAS). CM-Wh showed higher straw crop yield (21,104 $\pm$ 1,129 kg ha$^{-1}$), in comparison with the SM-Wh (16,909 $\pm$ 1,785 kg ha$^{-1}$). Both straw and root length were higher in the case of SM-Wh (Table 2). Nutrient's content (N, P, and K) was higher in the Wh with CM in comparison with the SM, except for Fe, which was three-fold higher in the SM.

## Effect of agricultural management on carbon and water fluxes

During the crop period (September 3rd, 2020, to April 27th, 2021), the environmental conditions were typical to this semi-arid region (Fig. S1). Global radiation was 228.72 $\pm$ 308.79 W m$^{-2}$ (mean $\pm$ standard deviation), relative humidity of 45.71 $\pm$25.05%, and air temperature 15.58 $\pm$ 7.92 °C, for a vapor pressure deficit (VPD) of 12.17 $\pm$ 10.17 hPa. The implementation of SM or CM influenced the C flux from the ecosystem to the atmosphere, particularly in the Np crop. Averages of GPP and Reco showed a similar tendency during the crop period (Fig. S2). As expected, the Np crops showed a lower daily mean GPP (1.85 $\mu$mol C m$^{-2}$ s$^{-1}$) and Rsoil (6.25 $\mu$mol C m$^{-2}$ s$^{-1}$) than the Wh crops, with GPP (6.34 $\mu$mol C m$^{-2}$ s$^{-1}$) and Rsoil (11.36 $\mu$mol C m$^{-2}$ s$^{-1}$) (Fig. 3). The SM-Np crop showed higher daily mean GPP, ET, and lower Reco. On the other hand, only daily mean ET was sensitive to management in the Wh crops with a higher ET in the CM (Fig. 3B).

Management had contrasting effects over total GPP and ET depending on the crop. SM-Np showed a GPP of 56.27 g C m$^{-2}$ and CM-Np of 21.92 g C m$^{-2}$, and an accumulated ET of 20.91 and 11.97 mm for SM-Np and CM-Np, respectively. Conversely, SM-Wh had a GPP of 184.05 g C m$^{-2}$ and CM-Wh of 331.79 g C m$^{-2}$. Total ET was also higher for the CM-Wh (125.16 mm) than the SM-Wh (65.93 mm; Fig. 4). These accumulated values also showed that SM influenced the capacity of the crops to use water, with higher WUE of the crops subjected to SM (1.42 and 1.03 g C m$^{-3}$ H$_2$O for Np and Wh, respectively) than the CM (0.77 and 0.48 g C m$^{-3}$ H$_2$O for Np and Wh, respectively), regardless of the crop (Fig. 4C).

According to the principal component analysis (Fig. 5), no difference was found between treatments since there was a similar individual distribution in the ordination. The VPD was the most important variable for the C and water flux in all crops and managements (Table S4). In this sense, the GPP was explained by the ET ($R^2 = -0.23$ for Np) in turn explained by the SWC ($R^2 = 0.37$ and 0.13 for Np and Wh, respectively). Reco was explained by Tair ($R^2 = 0.54$ and 0.55 for Np and Wh, respectively, Fig. S3).

**Table 1  Soil properties of nopal (Np) and wheat (Wh) crops subjected to conventional (CM) or sustainable (SM) management in the Chihuahuan desert.**

| | | Post-tillage | | | | Post-harvesting | | | |
| | | Nopal | | Wheat | | Nopal | | Wheat | |
| | Pre-tillage | CM (30 cm deep) | SM (manual) | CM (30 cm deep) | SM (10 cm deep) | CM | SM | CM | SM |
|---|---|---|---|---|---|---|---|---|---|
| *Organic matter (%)* | 4.45 (0.35) | 9.38*** (0.38) | 6.33*** (0.40) | 8.23 (0.34) | 7.61 (0.28) | 4.49 (0.09) | 4.45 (0.24) | 4.74*** (0.1) | 6.9*** (0.34) |
| *Organic carbon (%)* | 3.19 (0.1) | 7.19*** (0.06) | 5.36*** (0.06) | 6.19* (0.06) | 5.19* (0.06) | 3.21 (0.07) | 3.23 (0.05) | 3.65 (0.1) | 3.95 (0.17) |
| *Bulk density (g cm$^{-3}$)* | 1.07 (0.01) | 0.94*** (0.01) | 1.02*** (0.01) | 0.95* (0.01) | 0.99* (0.001) | 1.14 (0.01) | 1.15 (0.02) | 0.95*** (0.001) | 0.99*** (0.001) |
| pH | 8.01 (0.1) | 8.16 (0.11) | 7.97 (0.06) | 8.39* (0.03) | 8.23* (0.05) | 8.49 (0.03) | 8.51 (0.05) | 8.27** (0.04) | 8.56** (0.08) |
| *Electric conductivity (mS cm$^{-1}$)* | 274.24 (16.78) | 190.04 (14.21) | 184.1 (18.30) | 316.23* (17.83) | 384.04* (23.14) | 268.03 (19.24) | 346.24 (71.43) | 439.82 (18.41) | 521.6 (33.7) |
| *Available N (ppm)* | 1800 (0.1) | 2200 (10) | 2300 (10) | 2100 (10) | 2300 (10) | 1800 (10) | 1900 (10) | 1800** (10) | 2100** (10) |
| *Available P (ppm)* | 4.46 (0.24) | 4.68 (0.20) | 4.46 (0.17) | 5.10 (0.14) | 4.96 (0.17) | 2.21 (0.40) | 2.23 (0.32) | 1.22** (0.23) | 2.78** (0.44) |

**Notes.**

Data represent means (standard error) ($n = 12$). Asterisks indicate significant differences between managements for a given crop according to Tukey's *post hoc* comparison (significance level ***0.001,**0.01,*0.05). See Table S3.

**Table 2  Wheat yield and nutrient content under conventional (CM) or sustainable (SM) management in the Chihuahuan desert.**

| | | CM | SM |
|---|---|---|---|
| *Plant morphologic characteristics* | *Total biomass (g)* | 102.23* (3.78) | 82.56* (8.74) |
| | *Straw lenght (cm)* | 93.16 (7.34) | 102.58 (2.42) |
| | *Root lenght (cm)* | 17.41*** (1.39) | 26.08*** (1.37) |
| *Nutrients content* | *N (ppm)* | 9800*** (10) | 6700*** (70) |
| | *P (ppm)* | 600*** (10) | 400*** (10) |
| | *K (ppm)* | 16,100*** (20) | 10,400*** (10) |
| | *Fe (ppm)* | 149.01*** (5.07) | 456.5*** (102.1) |

**Notes.**

Data represent means (standard error) ($n = 12$ for yield and lengths and $n = 3$ for nutrients content). Asterisks indicate significant differences between managements for a given crop according to Tukey's *post hoc* comparison (significance level ***0.001,**0.01,*0.05). See Table S4.

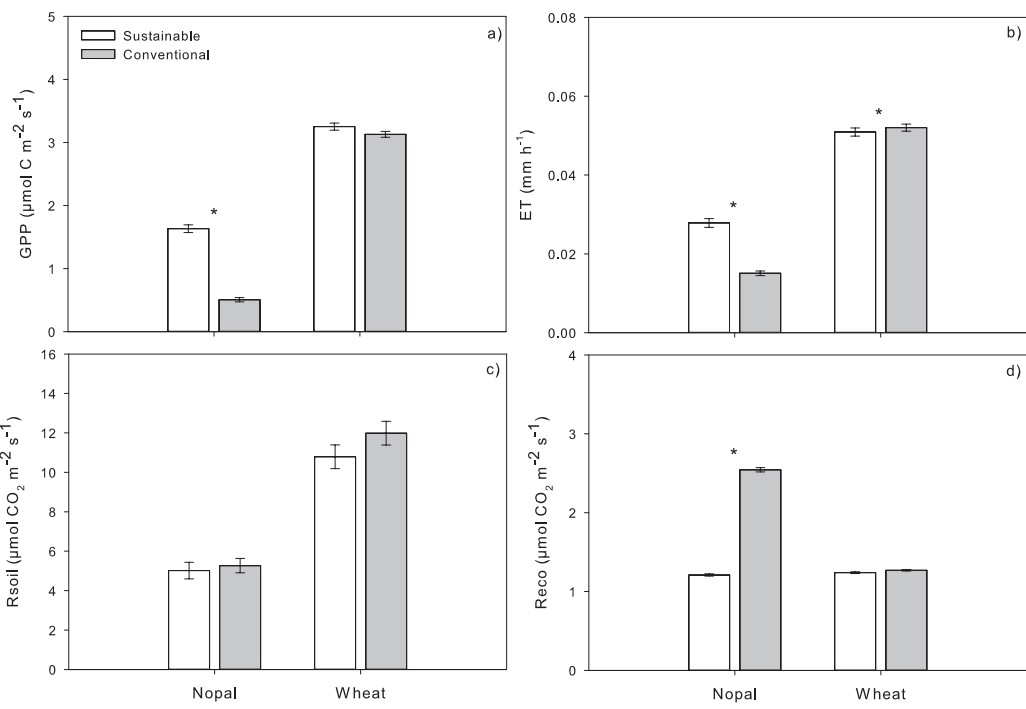

**Figure 3  Effect of agricultural management on gross primary productivity (GPP), Evapotranspiration (ET), ecosystem (Reco), and soil respiration (Rsoil) of nopal (Np) and wheat (Wh) crops subjected to conventional (CM in gray bars) or sustainable (SM in white bars).** Data represent means ±standard error ($n = 636$ for GPP, ET and Reco; $n = 240$ for Rsoil). Asterisks indicate significant differences between managements for a given crop according to Tukey's *post-hoc* comparison (significance level***0.001, ** 0.01,*0.05). See Table S1.

# DISCUSSION

## Agricultural management effect on soil characteristics and crop yield

The increase in soil organic matter (SOM) recorded in the plots post-tillage compared to the pre-tillage is due to the vegetation clearing, were fragments, both aerial and roots, were

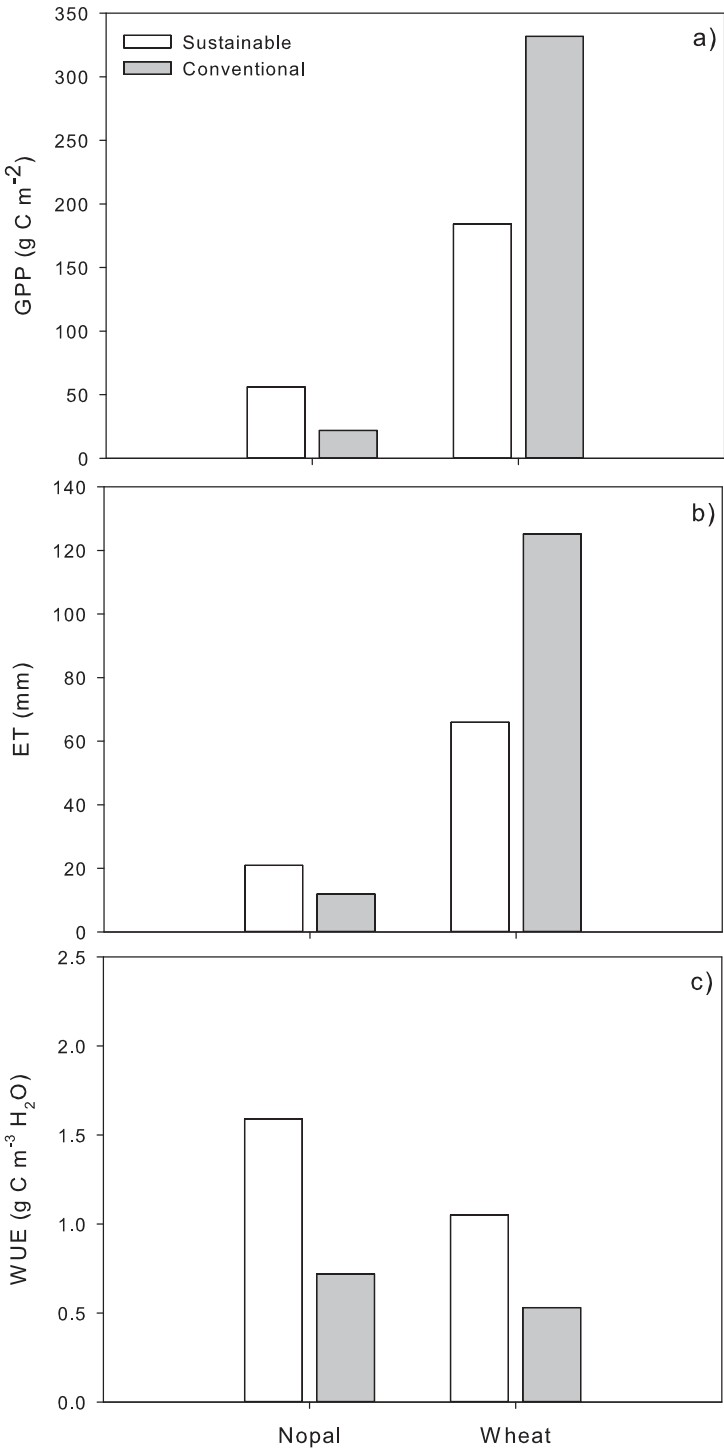

**Figure 4** (A) Total gross primary productivity (GPP), (B) evapotranspiration (ET), and (C) water use efficiency (WUE) of nopal (Np) and wheat (Wh) crops subjected to conventional (CM in grey bars) or sustainable (SM in white bars) management in the Chihuahuan desert.

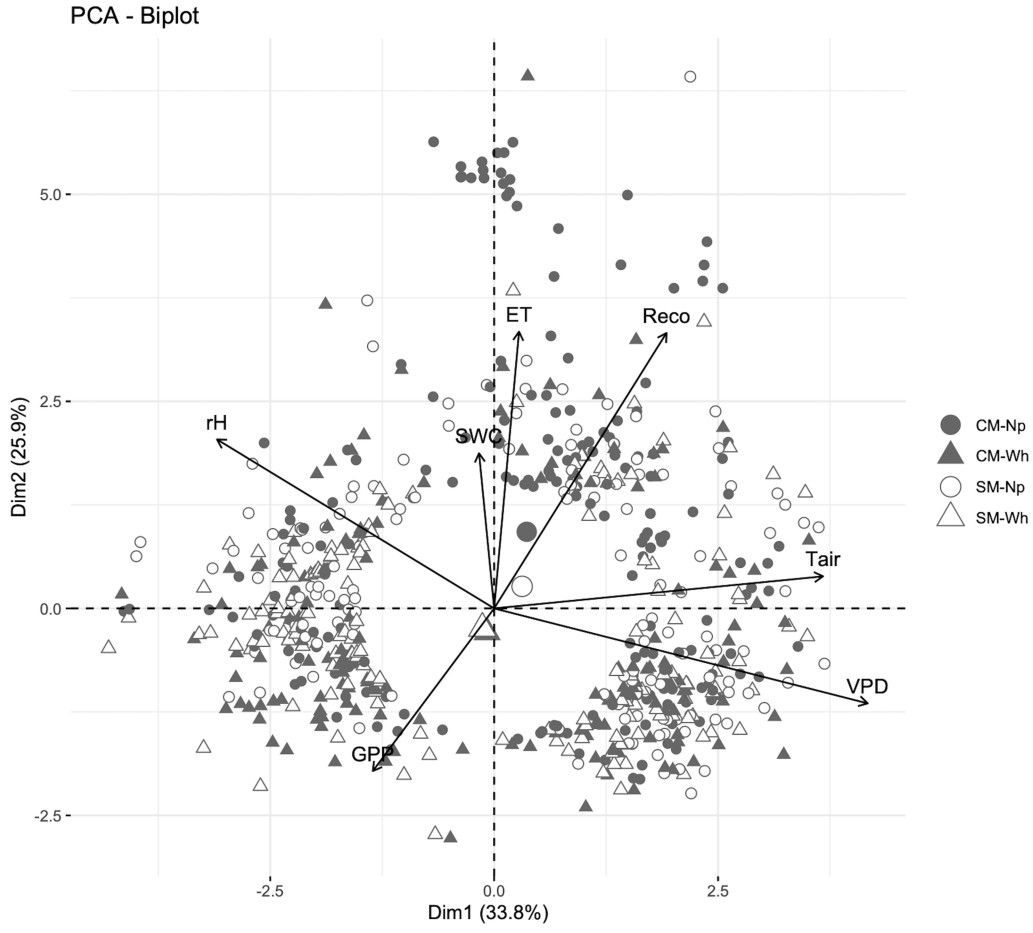

**Figure 5 Principal component analysis for nopal (Np) and wheat (Wh) crops subjected to conventional (CM) or sustainable (SM) management in the Chihuahuan desert.** GPP, gross primary productivity; $R_{eco}$, ecosystem respiration, SWC, soil water content, VPD, vapor pressure deficit, Tair, air temperature. Daily mean data, $n = 636$. See Table S2.

incorporated into the soil (*Ramesh et al., 2019*). Deeper tillage in the upper layer (0–30 cm) applied in the CM plots, regardless of the crop, results in a higher increase in SOM (4.4% more) than in the SM reduced tillage in the top layer (0–10 cm), which increases only 2.5% of SOM. This increase is explained by the mixing process that exposes the deeper SOM, potentially promoting its loss as $CO_2$ by decomposition in the middle and long term (*Post & Kwon, 2000*; *Haddaway et al., 2017*). In fact, after harvest, the SOM content is reduced in all plots due to natural decomposition processes, with a higher loss in CM plots (4.2%), compared to SM plots (1.3% of SOM loss). The SM-Wh presented the lower loss of SOM, with less than a 1% difference between post-tillage and post-harvesting determinations. The SOM pattern also reflects the input of compost in the SM management, and the crop plant material input, which is higher in the case of the Wh.

Furthermore, SOC, as part of the SOM composition (*Jackson et al., 2017*), has similar behavior. It has been consistently observed that the reduction of tillage intensity increases

the SOC concentration (*Liu et al., 2003*; *Singh et al., 2014*; *Haddaway et al., 2017*). Reduced tillage potentially produces benefits that result from soil C accumulation in the surface soil, such as improved infiltration, water-holding capacity, nutrient cycling, soil biodiversity, and erosion reduction (*Busari et al., 2015*; *Haddaway et al., 2017*). Likewise, in this study, tillage influenced bulk density, soils with SM have higher bulk density than CM ones, as reported in other agricultural studies (*McVay et al., 2006*; *Jat et al., 2018*).

Irrigation and fertilization tend to increase the pH and the electrical conductivity in agricultural fields (*Zhao et al., 2007*; *Darvishi, Manshouri & Farahani, 2010*); in this study, we observed such changes except in the CM-Wh. Specifically, the soil pH decreases in the CM-Wh post-harvesting can be associated with the fertilizer applied, since urea releases $H^+$, resulting in slight soil acidification (*Hao & Chang, 2003*; *Tian & Niu, 2015*). The significant increase in electrical conductivity in soil from SM, regardless of the crop, is explained by the compost application, which by presenting $Na^+$ and $Mg^+$ salts, increases soil salinity (*Artiola et al., 2019*); however, it is still lower than the critical level of 4 dS $m^{-1}$ (*Jat et al., 2018*). The control of these variables is of utmost agricultural importance because these control the bioavailability of nutrients such as N, P, and K, affecting also microbial activity (*Heiniger, McBride & Clay, 2003*; *Xue et al., 2018*).

Available N and P content in the soil showed increases after tillage, indicating the incorporation of fertilizers (*i.e.*, urea for CM and compost for SM) and a decrease after harvesting due to its incorporation and metabolization by the plant and soil microorganisms. However, the nutrient reduction was higher in CM than SM plots, and higher for Wh than Np. The last can be explained by the inherent slow metabolism of the Np, with lower nutrient requirements (*Consoli, Inglese & Inglese, 2013*). Furthermore, the application of compost as fertilizer in the SM results in a higher increase of soil N with lower loss in the soil, and lower incorporation by the crop, specifically the wheat, negatively impacting the crop yield.

Wheat nutrient uptake was higher in the CM crop than in the SM crop, resulting in higher biomass production and yield. The poor wheat nutrient uptake by the SM management can be attributed to the tillage that affected the root growth of this management (*Muñoz Romero et al., 2010*), which did not penetrate deep enough for better development (*Mašková & Herben, 2018*), and to the application of N in a less available form (compost). The lower N availability reduced biomass production (*Litke, Gaile & Ruža, 2018*). Furthermore, N allows proper uptake of the remaining micronutrients, thus causing a higher nutrient deficit in the crop (*Cakmak, Pfeiffer & McClafferty, 2010*; *Hamnér et al., 2017*; *Singh, 2019*). The crop yield was similar to previously reported for this crop during winter growth (*Curtin et al., 2000*; *Bista et al., 2017*). The differences in production between crops are associated with the slow development that Np presented during growth, compared to other regions, due to the low temperatures and the limited irrigation provided (*López Collado et al., 2013*). Contrary to the reported nopal production in other semi-arid areas, the cladode growth in our case did not reach the minimum size to be cultivated (*Liguori et al., 2013*; *López Collado et al., 2013*; *Snyman, 2013*). Despite the advantages offered by the Crassulacean Acid Metabolism species in terms of productivity and resistance to drought

over conventional semi-arid crops $C_3$ and $C_4$, the low yield we obtained did not allow us to perform a comparison (*Owen et al., 2016*).

**Agricultural management effect on carbon and water flux**

Management practices in agriculture modify the dynamic in the C, N, and water fluxes (*Davis et al., 2010*; *Fisher et al., 2017*; *Camarotto et al., 2018*; *Mirzaei et al., 2021*; *Mirzaei & Caballero Calvo, 2022*), specifically, sustainable management has the potential to reduce the C flux from the soil to the atmosphere. This is especially relevant considering that arid zones' current and future C sink capacity will strongly depend on water availability (*Hovenden, Newton & Wills, 2014*). Winter wheat crops managed with a reduced tillage decrease the ecosystem respiration (Reco) (*Bista et al., 2017*) compared to conventional tillage. Our study found no significant differences between managements in both daily mean carbon uptake (GPP) and release (Reco and Rsoil) for the Wh crop, although a tendency of this decrease in Rsoil and increase in GPP was detected. The lack of differences can be explained by the length of the study since the application of the SM practices had more impact in the long term (*Busari et al., 2015*; *Haddaway et al., 2017*). However, in the case of Np, SM significantly reduced the Reco while significantly increased the GPP in comparison with CM. Furthermore, the total carbon uptake (GPP) was larger than the release (Reco) on the SM plots regardless of the crop, whereas the CM plots behave as net C source, as has been previously described for different crops (*Curtin et al., 2000*; *Vote, Hall & Charlton, 2015*; *Bista et al., 2017*; *Heimsch et al., 2021*).

Furthermore, the differences in crops metabolism resulted in lower C uptake by Np. While the wheat (*T. aestivum*) has a $C_3$ metabolism, the $CO_2$ fixation rate *via* Rubisco is higher than the Malate route used in the CAM of the nopal (*O. ficus*) (*Bhagwat, 2005*; *Snyman, 2013*). These differences in metabolisms also produce significant differences in crop water use. While the Wh $C_3$ metabolism does not have photosynthetic adaptations to reduce photorespiration, the CAM metabolism of the Np prevents photorespiration during the day, thus increasing WUE (*Consoli, Inglese & Inglese, 2013*; *Liguori et al., 2013*; *Owen et al., 2016*). As a result, the Wh crops exhibit a higher ET than Np. As we first hypothesized, the historical physiological adaptations presented by the native Np result in a CAM metabolism that allow it to have lower water requirements and ET and, therefore, a higher WUE than Wh (*Borland et al., 2009*; *Guillen-Cruz et al., 2021*).

On the other hand, the management also influenced the total WUE, the SM (*i.e.*, minimum tillage, compost) for both Np (drip-irrigated) and Wh (micro-aspersion), in which the SWC was reduced (by 3% and 7% than the CM), over time the SWC present a more stable behavior, showing that SM improved the WUE of crops by reducing ET (*Zheng et al., 2018*), being especially noticeable in Wh.

## CONCLUSIONS

Differences in the management of wheat and nopal changed the soil quality and the carbon and water fluxes. Although the use of C by nopal was lower due to metabolic differences with wheat, the management effect on the C and water cycles was remarkable. Sustainable management (*i.e.*, reduced tillage and organic fertilization) showed more efficient water

use in incorporating C into the plant, also showing higher soil water retention and a lower loss by evapotranspiration in comparison with conventional management (*i.e.*, regular 30 cm tillage and urea fertilization). Sustainable practices enhance and maintain soil quality (change in organic matter and nutrient content). However, it results in a lesser nutrients plant uptake, so fertilization must be fine tuned to ensure proper plant nutrition within the range of sustainable possibilities.

## ACKNOWLEDGEMENTS

The authors thank Andres Torres-Gómez, Emmanuel F. Campuzano, Agustin Torres and Cesar Sarabia-Castillo for their help in the experimental implementation.

### Funding

This work was supported by the Fondo Destinado a Promover el Desarrollo de la Ciencia y la Tecnología en el Estado de Coahuila, del Consejo Estatal de Ciencia y Tecnología de Coahuila (COAH-2020-C14-C091). Roberto Torres-Arreola and Gabriela Guillen Cruz received M.Sc. (755725) and Ph.D. (779550) scholarships from CONACyT, respectively. The funders had no role in study design, data collection and analysis, decision to publish, or preparation of the manuscript.

### Grant Disclosures

The following grant information was disclosed by the authors:
FONCYT: COAH-2020-C14-C091.
CONACyT:  M.Sc. (755725),  Ph.D. (779550).

### Competing Interests

The authors declare there are no competing interests.

### Author Contributions

- Gabriela Guillen Cruz conceived and designed the experiments, performed the experiments, analyzed the data, prepared figures and/or tables, authored or reviewed drafts of the article, and approved the final draft.
- Roberto Torres-Arreola performed the experiments, analyzed the data, prepared figures and/or tables, authored or reviewed drafts of the article, and approved the final draft.
- Zulia Sanchez-Mejia analyzed the data, prepared figures and/or tables, authored or reviewed drafts of the article, and approved the final draft.
- Dulce Flores-Renteria conceived and designed the experiments, performed the experiments, analyzed the data, prepared figures and/or tables, authored or reviewed drafts of the article, and approved the final draft.

### Data Availability

The raw data is available at Zenodo: Gabriela Guillén Cruz, Roberto Torres Arreola, Zulia Sanchez-Mejia, & Dulce Flores-Renteria. (2022). Data set [Data set]. Zenodo. https://doi.org/10.5281/zenodo.7076322.

## Supplemental Information

Supplemental information for this article can be found online at http://dx.doi.org/10.7717/peerj.14542#supplemental-information.

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
