# Peer review of "The effect of conventional and sustainable agricultural management practices on carbon and water fluxes in a Mexican semi-arid region"

_PeerJ, doi:10.7717/peerj.14542_

## Round 0.1 · original submission · Minor Revisions

Please revise according to the comments of reviewers.

Reviewer 1 ·

Basic reporting

This work is very interesting and the contribution essential in the field of sustainable agriculture. The wording is very clear. The literature review is deep and up to date, maybe these papers could help to interpret the results:

Snyman, H. A. (2013). Growth rate and water-use efficiency of cactus pears Opuntia ficus-indica and O. robusta. Arid Land Research and Management, 27(4), 337-348. https://doi.org/10.1080/15324982.2013.771232

Owen, N. A., Choncubhair, Ó. N., Males, J., del Real Laborde, J. I., Rubio‐Cortés, R., Griffiths, H., & Lanigan, G. (2016). Eddy covariance captures four‐phase crassulacean acid metabolism (CAM) gas exchange signature in Agave. Plant, Cell & Environment, 39(2), 295-309. https://doi.org/10.1111/pce.12610

Figures symbology could be improved, please consider using triangles for wheat (gray for CM, white for SM) and circles for nopal.

Raw data is available in according to the Data Sharing Policy.

Experimental design

Although the objective and the hypothesis are clearly presented, it is necessary to elaborate why is important to compare wheat and nopal from the perspective of profitability in agricultural production, environmental or other points of view. If there is not a robust justification to compare these crops, the main objective must emphasize the comparison between management practices (Sustainable and conventional).
The results of the work are consistent with the hypothesis and the methodology used.

The experimental design is well done and the results are correct. The methods are well described, but it is necessary to detail some procedures:
1) Please explain how aerial wheat biomass was measured?
2) Regarding the soil properties comparison in post-tillage, please explain why the samples were separated into crops? The crops were established before the sampling?.
3) Please detail, is there any consideration in partitioning GPP and Reco due to different crop metabolisms?
4) How do you determinate the crop water requirements? Since the SWC was maintained in a different level for each crop and management type.

Validity of the findings

Please verify values of accumulated ET for wheat in figure 4.

L336 " In the case of the Nopal (Np), the yield was lower than the reported (Liguori et al. 2013; López Collado et al. 2013)" How do you know that? Please explain why a destructive biomass quantification for Nopal crop was not performed.

In figure 3 footprint of 40 meters is shown. If experimental plot radius was of 10 m from de EC tower, which could be the influence of the surrounding vegetation?

Please report what proportion of the Eddy Covariance flux data remained after quality control procedure.

Please explain post-tillage conditions. If there were no crops at measurement moment, a statistic test comparing management type (CM and SM) and pre-tillage conditions could improve the results presentation.

Additional comments

I think the main title could be improved removing "of contrasting crops"

Figure 2. Please mark with the letters a and b in the image.

L111 "The study area was delimited into four 20x20 m plots." In Figure 1 each plot has 10x10 m, please verify.

L124 change " on early September " for "in early September"

Please clarify if in this sentence nopal values are respectively for SM and CM: "SWC was maintained for Wh between 17±5% and in 8±4% by applying between 5-20 mm3 for SM and 15-45 mm3 for CM, and for Np between 8±5% and in 5±4%."

L150-151 "Wh crop was characterized by taking five cores of the plant crop (4” diameter) at each plot." Is this sentence referring to evaluate biomass production? If this is true, why did you use cores instead a standard forage cuttings?

L207 Remove parenthesys (<= 40 m

L215 Please "21.5 from" add % symbol.

L314 to 316. This sentence needs a reference. "The control of these variables is of utmost agricultural importance because these controls the bioavailability of nutrients such as N, P, and K, affecting also microbial activity".

L272-273 "Total ET was also higher for the CM-Wh (125.16 mm) than the SM-Wh (65.93 mm; Fig. 4)." These values do not match in figure 4, please verify.

L334-336. Precipitation is not the limiting factor considering that crop was irrigated.

L374 I think conclusion can be delimit to specific management practices applied (Tillage and fertilization), because the effect of all management practices was not explored.

Reviewer 2 ·

Basic reporting

'no comment

Experimental design

'no comment

Validity of the findings

'no comment

Additional comments

The authors evaluated “Conventional and sustainable agricultural management effect of contrasting crops on carbon and water fluxes in a Mexican semi-arid region”. The work is interesting and I consider the undertaken research interesting from a scientific and practical point of view for publication. However, it need some minor revisions to be improved. I wrote some suggestions in the main text. Also, the manuscript need the English checking.

Annotated reviews are not available for download in order to protect the identity of reviewers who chose to remain anonymous.

---

## Round 0.2 · accepted · Accept

The authors have addressed all of the reviewers' comments.